# Tryptophan Metabolism and Gut-Brain Homeostasis

**DOI:** 10.3390/ijms22062973

**Published:** 2021-03-15

**Authors:** William Roth, Kimia Zadeh, Rushi Vekariya, Yong Ge, Mansour Mohamadzadeh

**Affiliations:** 1Department of Neurology, University of Florida, Gainesville, FL 32608, USA; 2Department of Infectious Diseases & Immunology, University of Florida, Gainesville, FL 32608, USA; kimmmayy1@ufl.edu (K.Z.); rvekariya@ufl.edu (R.V.); gey@ufl.edu (Y.G.); 3Division of Gastroenterology, Hepatology & Nutrition, Department of Medicine, College of Medicine, University of Florida, Gainesville, FL 32608, USA

**Keywords:** tryptophan, serotonin, kynurenine, kynurenic acid, quinolinic acid, gut microbiota, inflammation, cerebrovascular disease, anxiety, depression, inflammatory bowel disease

## Abstract

Tryptophan is an essential amino acid critical for protein synthesis in humans that has emerged as a key player in the microbiota-gut-brain axis. It is the only precursor for the neurotransmitter serotonin, which is vital for the processing of emotional regulation, hunger, sleep, and pain, as well as colonic motility and secretory activity in the gut. Tryptophan catabolites from the kynurenine degradation pathway also modulate neural activity and are active in the systemic inflammatory cascade. Additionally, tryptophan and its metabolites support the development of the central and enteric nervous systems. Accordingly, dysregulation of tryptophan metabolites plays a central role in the pathogenesis of many neurologic and psychiatric disorders. Gut microbes influence tryptophan metabolism directly and indirectly, with corresponding changes in behavior and cognition. The gut microbiome has thus garnered much attention as a therapeutic target for both neurologic and psychiatric disorders where tryptophan and its metabolites play a prominent role. In this review, we will touch upon some of these features and their involvement in health and disease.

## 1. Tryptophan

### 1.1. Dietary Intake and Absorption

Tryptophan (TPH) is an essential amino acid found primarily in proteinaceous food such as milk, tuna, turkey, oats, cheese, nuts, and seeds [1,2]. TPH availability is thus largely dependent on dietary intake. Several factors influence circulating TPH levels, including cultural and geographic dietary practices, stress, and physical activity [3,4]. Formula-based diets for infants may provide less tryptophan than breast milk, which itself varies considerably in concentration based on cultural and geographic dietary practices [5].

The vast majority of TPH is absorbed in the small intestine [6]. Concomitant host ingestion of other neutral amino acids limits TPH absorption and availability. Similarly, dietary carbohydrates reduce TPH uptake as induction of insulin release stimulates preferential capture of non-TPH neutral amino acids [7]. Once in the peripheral circulation, TPH is 80–90% albumin-bound and the rest is freely solubilized. TPH enters the central nervous system (CNS) via the L-type amino acid transporter (LAT1/Lat1) responsible for shuttling large neutral amino acids across the blood-brain barrier (BBB) [8]. This protein complex antiporter is heavily expressed in astrocytic foot processes that comprise the BBB. Further, investigations in murine models suggest that Lat1 is expressed by neurons, astrocytes and microglia, whose cellular frequencies comprise a secondary barrier for central TPH uptake [9]. The unabsorbed TPH in the small intestine also serves as a metabolic substrate for colonic microbes, which will be discussed in detail below.

### 1.2. Host Metabolic Pathways

#### 1.2.1. Serotonin Pathway

TPH serves as the only substrate for serotonin synthesis (Figure 1), which occurs primarily in the distal gastrointestinal tract (90%) and, to a lesser extent, in the central nervous system (10%) [10]. Tryptophan hydroxylase catalyzes the initial, rate-limiting step in the conversion of tryptophan to serotonin. Within the gut, enterochromaffin cells (ECs) express tryptophan hydroxylase (TPH1), while serotonergic neurons express TPH2 in the central nervous system (CNS) and enteric nervous system (ENS). TPH1 and 2 convert TPH to the intermediate L-5-hydroxytryptophan (5-HTP) and L-amino acid decarboxylase converts 5-HTP to 5-hydroxytryptamine (5-HT), or serotonin. Within the pineal gland, TPH is converted by TPH1 to serotonin, which can be further converted into melatonin, the major endogenous regulator of sleep initiation and circadian rhythms. Serotonin is otherwise catabolized by monoamine oxidase (MAO) into 5-hydroxyindole acetaldehyde and then by aldehyde dehydrogenase to 5-hydroxyindole acetic acid (5-HIAA) that is excreted in the urine [11].

#### 1.2.2. Kynurenine Pathway

TPH can further be oxidized via the kynurenine degradation pathway. Accordingly, approximately 90% of the TPH degraded in this fashion occurs in the liver, wherein tryptophan 2,3-dioxygenase (TDO) performs the initial rate-limiting step. The remaining TPH is initially degraded through this pathway in the brain, GI tract, and liver by indoleamine 2,3-dioxygenase (IDO). Proinflammatory cytokines, IFNγ in particular, potently induce IDO activity, while glucocorticoids induce TDO expression. Kynurenine is further catabolized into two neuroactive inflammatory mediators, kynurenic acid (KA), and quinolinic acid (QA). Central kynurenine catabolites are produced locally and to a lesser extent cross the blood-brain barrier. In the brain, they act at neuronal N-methyl-D-aspartate (NMDA) receptors, glutamate-activated post-synaptic calcium ion receptors, which are important in learning and memory and overactivated in neuronal excitotoxicity. KA acts as a NMDA receptor antagonist at the glycine binding site and is generally regarded as a neuroprotective factor at physiologic concentrations, while QA, produced primarily in microglia, acts as a weak agonist and is neurotoxic, owing to this effect and its enhancement of free radical production [12]. KA also potently antagonizes α7 nicotinic receptors throughout the body. QA is further catabolized into niacin and NAD^+^, which are active in other critical cellular metabolic processes. Of note, vitamin B6 (VB6) operates as a cofactor in several critical pyridoxal 5′-phopsphate-dependent catabolic pathways. As a result, VB6 deficiency results in derangements in catabolite levels [13].

## 2. Tryptophan Metabolites

### 2.1. Serotonin

#### 2.1.1. Homeostasis

Enterochromaffin cells (ECs) in the intestinal mucosa synthesize the vast majority of serotonin in the body. Dietary tryptophan is converted via TPH hydroxylase 1 (TPH1) to serotonin in secretory granules primarily at the basal surface and to a lesser extent at the apical surface of ECs. When activated, ECs release serotonin into the interstitium of the lamina propria where it binds nerve endings of intrinsic sensory neurons. Serotonin cellular uptake for degradation by MAO occurs via the selective serotonin reuptake transporter (SERT), which all intestinal epithelial cells express, allowing for highly efficient reuptake shortly after EC secretion. Platelets circulating in the dense capillary network of the lamina propria take up remaining serotonin via SERT. Importantly, platelets serve as a systemic reservoir of serotonin and can deliver it to remote peripheral tissue. Gut microbes influence serotonin production and secretion through cell- and metabolite-mediated processes [14].

#### 2.1.2. Gut Signaling

Gut intrinsic serotonergic neurons comprise approximately 2% of all enteric neurons, and similar to central neurons, express TPH2 for tryptophan conversion to serotonin. These neurons typically form excitatory synaptic connections with other serotonergic neurons in the intestine where they account for about 10% of total excitatory activity, while the majority occurs by cholinergic activity at nicotinic receptor sites. Available data suggests that these neurons exert downstream effects from descending pathways and mediate circular muscle activity, while some other data suggests afferent activity. Moreover, differential electrophysiological effects appear to be mediated by serotonin receptor subtype expression by target cells [15,16].

#### 2.1.3. Gut Motility

Intestinal intraluminal pressure and mechanical stimulation of ECs activate serotonin release, which in turn elicit the activation of the neural-mediated contraction and relaxation arcs of the peristaltic reflex. 5-HT3 and 5-HT4 receptors appear to mediate gut motility as antagonism of these receptors results in reduced motility in humans and animal models [17]. Further, 5-HT3 antagonists such as ondansetron possess potent antiemetic effects and are used as first-line agents for the treatment of nausea [18]. Notably, prolonged exposure to agents that increase the interstitial availability of serotonin cause receptor desensitization, which leads to reduced colonic transport [19]. Serotonin does not appear to be solely responsible for initiation of peristalsis, as mechanical stimulation induces peristalsis in TPH1-deficient mice [20]. Serotonin, in concert with cholecystokinin, orchestrates colonic segmentation in the small intestine, primarily by its action at the 5-HT4 receptors [16,21]. Additionally, serotonin plays a major role in the secretomotor reflex, where it activates sensory afferent nerve endings expressing 5-HT3 and 5-HT4 receptors in the lamina propria, inducing submucosal ganglionic cholinergic and vasoactive intestinal peptide (VIP)-producing neuronal activity. Neuronally-activated secretory epithelial cells then release chloride and bicarbonate anions, neutralizing luminal contents and protecting against luminal pathogens. Serotonin may also induce the activation of intestinal epithelial cells to secrete this substance through paracrine action. Serotonin-associated activation of 5-HT3 and 5-HT4 receptors concomitantly stimulates a local neutrally-mediated vasodilatory response. Interstitial cells of Cajal (ICCs) are widely-distributed “pacemakers” that regulate spontaneous gut rhythmic motility. Serotonin promotes ICC network survival in a 5-HT2B receptor-dependent process. 5-HT2B-deficient mice demonstrate an increased ICC density and impaired spontaneous rhythmic gut motility [22].

Serotonin also stimulates vagal nerve endings in the upper GI tract expressing the 5-HT3 receptor. Infusion of luminal glucose causes serotonin release from ECs that stimulate a vagal reflex arc, which delays gastric emptying and activates pancreatic insulin secretion. Luminal lipid infusions promote satiety via 5-HT3- and CCK1-dependent vagal signaling, and 5-HT3 antagonism contributes to greater food intake by humans [23]. Serotonin also appears to play a major role in abdominal pain signaling, as 5-HT3 antagonism ameliorated pain behavior and dorsal horn activation induced by noxious colonic distension in a murine model [24,25]. In addition to its impact on vagal afferent activity, activation of parasympathetic neurons causes elevations in circulating serotonin [24].

#### 2.1.4. Vasoreactivity

Serotonin is a major regulator of vasoreactivity that elicits differential responses based on the target vasculature. In vivo models demonstrate that serotonin directly induces vasoconstriction in large arteries and veins and enhances the contractile effect of other vasoconstrictors such as angiotensin II and histamine [26,27,28]. It exerts a vasodilatory effect in arterioles via 5-HT1 receptor-activated nitric oxide release and vascular smooth muscle relaxation. Central serotonergic neurons further influence vasomotor tone via 5-HT1 and 5-HT2 receptor activation [29].

#### 2.1.5. Platelet Activation

Platelets, in addition to storage and trafficking of serotonin, express serotonin receptors that, when stimulated, promote platelet aggregation. While serotonin alone weakly activates platelets, in the presence of other platelet activators, including thromboxane A2 and adenosine diphosphate, it potentiates this activity in a 5-HT2 receptor-dependent process [30].

#### 2.1.6. Immune Cell Activity

Several immune cells express serotonin receptors, including B and T lymphocytes, monocytes, macrophages, and dendritic cells (DCs) suggesting an immunomodulatory role for serotonin [31]. Subclasses of the 5-HT1 and 5-HT2 receptors are expressed by B and T lymphocytes, monocytes, macrophages and DCs. However, less is known about the expression patterns of other serotonin receptors by immune cells, although recent work by Ito and colleagues [32] illustrates the important role of 5-HT7 receptor in brain regulatory T cell function after ischemic stroke. Immune cells also serve as regulators of circulating serotonin levels by phagocytic cells (e.g., DCs), and lymphocyte uptake through the 5-HT transporter (5-HTT). In human DCs and peripheral blood monocytes (PBMCs), serotonin stimulates production of pro-inflammatory IL-1β, while decreasing TNFα in these cells [33]. It further acts to stimulate IFNγ release by natural killer cells and IL-16 in circulating leukocytes, especially CD8^+^ T cells [34]. Additionally, serotonin receptor families mediate differential cytokine responses. 5-HT2A activation inhibits TNFα by PBMCs and 5-HT4 and 5-HT7 acts in the same capacity on monocytes. In DCs, 5-HT4 and 5-HT7 activation increases IL-1β and IL-8 while reducing IL-12 and TNFα [35]. Moreover, activation of 5-HT3 receptors also stimulate IL-1β and IL-6 production [36]. Furthermore, systemic effects of serotonin activation on peripheral tissues include bronchoconstriction, regulation of bone density, stimulation of hematopoiesis and nociception [29].

#### 2.1.7. Central Serotonin Activity

The hypothalamus-pituitary-adrenal (HPA) axis has emerged as an important pathway within the microbiota-gut-brain axis. In pediatric patients experiencing parental separation, HPA axis activation has been linked to changes in serotonergic function. Here, 5-HT1A and 5-HT1B receptor expression is reduced throughout the brain [37]. Further, maternally-separated rats have increased detectable serotonin in the spinal cord and colon and an increase in intestinal EC density following induced colonic distension. Further, serotonergic neurons in the dorsal raphe nucleus responsible for top-down pain inhibition are highly-activated in these rats [38,39,40,41,42].

Central serotonergic neuronal cell bodies reside in several functionally and anatomically distinct nuclei spanning the rostral-caudal axis of the brainstem, collectively referred to as the raphe nuclei [29]. These are categorized into rostral and caudal segments, influencing cerebrum, rostral brainstem and caudal brainstem, spinal cord and peripheral nerve signaling, respectively. Reciprocal projections between the serotonergic neurons of the raphe nuclei are ubiquitous and project to virtually the entire neuroaxis. The most robust signaling pathways primarily involve limbic forebrain structures, including the nucleus accumbens, the so-called “pleasure center” of the brain, the hypothalamus, substantia nigra and the periaqueductal gray involved in central inhibition of peripheral nociception. As opposed to a purely excitatory or inhibitory role, serotonin operates as a neuromodulator with variegated effects on target cells. Signaling occurs by both canonical synaptic transmission and by volume transmission, wherein the neurotransmitter diffuses beyond the synaptic cleft and can modulate nearby dendrites, adding an additional layer of complexity to understanding the physiology of serotonin in the functional circuitry of the brain. An understanding of the major physiological role of central serotonin has been gleaned through analysis of the function of serotonin receptor subtypes on target cells. Thus, abundance of serotonin receptors in the limbic system illustrates the major role of serotonin in regulating mood, cognition, pain, sleep, and neuroendocrine machineries. Serotonin receptors on large intracranial vessels mediate intracranial vasoconstriction. Receptor activation in the neocortex is primarily involved with integration of afferent sensorimotor data. 5-HT2C receptors are highly expressed on choroid plexus and facilitate CSF secretion when activated (Table 1) [43,44].

**Table 1 ijms-22-02973-t001:** Serotonin Receptor Subtypes. Serotonin receptors vary in expression in target tissues and are implicated in a variety of disorders [45,46].

Receptor Subtype	Primary Locations	Implicated Disorders
5-HT1A	CNS (Median raphe nuclei; septal nuclei; hippocampus; neocortex)	Anxiety and depression
5-HT1B	CNS (Ventral pallidum; substantia nigra), intracranial vasculature	Migraine
5-HT1D	CNS (Ventral pallidum; substantia nigra), intracranial vasculature	Migraine
5-HT1E	CNS (Olfactory bulb; hippocampus; neocortex; striatum)	?
5-HTF	CNS (Trigeminal ganglia; neocortex; hippocampus; astroglia), PNS (Dorsal root ganglia), Renal proximal tubule, coronary artery, pulmonary artery	Migraine
5-HT2A	CNS (Neocortex); vascular smooth muscle	Psychosis, schizophrenia, hypertension
5-HT2B	Cardiac Fibroblasts, stomach	Heart failure, anxiety, pulmonary hypertension
5-HT2C (formerly 5-HT1C)	CNS (Choroid plexus)	Obesity, epilepsy, psychosis, mood disorders, anxiety
5-HT3	CNS (Substantia gelatinosa; area postrema)	Nausea and vomiting
5-HT4	CNS (Hippocampus; colliculi), GI tract	GI motility disorders
5-HT5	CNS (Hippocampus)	Sleep disorders
5-HT6	CNS (Striatum; neocortex; limbic system)	Cognitive disorders, obesity, seizures
5-HT7	CNS (Suprachiasmatic nucleus; hippocampus; thalamus)	Anxiety, sleep disorders, cognitive disorders

### 2.2. Kynurenine

#### 2.2.1. Central Activity of Kynurenine Metabolites

Kynurenine is the main bioactive product of TPH catabolism. It crosses the BBB by active transport through large neutral amino acid carriers where it can be converted to KA and QA and thereby exert its downstream effects on neural signaling. Kynurenine metabolites, including KA, QA and anthranilic acid, from the systemic circulation can freely diffuse across the BBB. Kynurenine aminotransferases (KATs), present in cells of monocytic lineage, catalyze the conversion of kynurenine to KA [47]. Astrocytes preferentially metabolize kynurenine through the KA arm of the pathway as they lack kynurenine 3-hydroxylase which catalyzes the rate-limiting step in the QA arm of kynurenine degradation. However, activated microglia express these enzymes shunting kynurenine to the QA arm of the degradation pathway [48,49]. MicroRNAs (miRNAs), specifically 294-5-p, Brd2, and Slit3kr critically regulate kynurenine pathway enzymes [4,50]. Interestingly, hippocampi of GF mice exhibit elevated miRNAs, which can be normalized following microbial colonization in these mice, highlighting the critical role of gut microbiota [51].

#### 2.2.2. Kynurenic Acid

In addition to its role in NMDA receptor antagonism, KA potently antagonizes central and peripheral nicotinic α-7 receptors (nAch7Rs) [52]. These receptors are highly expressed on macrophages and lymphocytes. Vagal stimulation attenuates TNFα release by macrophages in the reticuloendothelial system, an effect mediated by nA7chRs, as this response is eliminated in nAch7R KO mice. Acetylcholinesterase inhibitor treatment also reduces TNFα, IL-1β, IL-6 and IL-8. Immune activation by microbial components such as lipopolysaccharide (LPS) and lipoteichoic acids (LTA) potently stimulate kynurenine degradation [53]. This effect appears to be mediated by activation of certain Toll-like receptors (TLRs), including TLR2, 3, 4, 7/8, and 9 [54]. Furthermore, microbial short chain fatty acids (SCFAs), particularly butyrate, alters IDO expression, thereby reducing kynurenine production. Butyrate affects the suppression of STAT1, which subsequently inhibits IFNγ-dependent STAT1 phosphorylation and STAT1-driven IDO activity [55].

#### 2.2.3. Quinolinic Acid

QA and its metabolites 3-hydroxyanthranilic acid (3-HANA) and 3-hydroxykynurenine (3-HK) exhibit tolerogenic properties putatively by blocking the activation and proliferation of B, T, and NK cells. Immature DCs expressing both IDO and QA metabolic enzymes significantly prevent autoimmunity in the hosts [56]. Other evidence by Mohamadzadeh and colleagues demonstrates that dietary TPH may induce autoimmunity in lupus-prone mice [57]. Further, immune stimulation of QA metabolic enzyme-producing macrophages and microglia enhance the QA metabolic pathways [58,59]. Kynurenine itself can activate colonic aryl hydrocarbon receptors (AHRs), thereby tuning epithelial immune regulation [60].

### 2.3. Gut Microbial Tryptophan Metabolism

Preclinical investigations have now established that differences in gut microbial composition and diversity can alter tryptophan availability in association with alterations in CNS development.

#### 2.3.1. Studying Tryptophan Metabolisms in Animal Models

Studies of animals devoid of a gut microbiome provide important evidence of the impact of the gut microbiome on tryptophan metabolism. Germ free (GF) mice exhibit increased plasma tryptophan and serotonin, which normalize following introduction of gut microbiota post-weaning [6]. Studies in GF mice revealed the complexity of tryptophan and serotonin availability in brain areas in which serotonin signaling plays a significant role. Accordingly, GF mice appear to demonstrate reduced TPH catabolism through the kynurenine degradation pathway compared to conventional mice. These GF mice have higher circulating TPH and lower kynurenine. Further, GF mice showed that males have elevated hippocampal serotonin that persists following the reconstitution of the gut microbiome and associated normalization of circulating tryptophan concentrations [61]. Although, another study employing GF rats showed decreased hippocampal and medial prefrontal serotonin. These rats did, however, demonstrate a robust stress-induced elevation of serotonin and 5-HIAA and serotonin in frontal cortical and hippocampal tissue. Conflicting findings in brain serotonin levels in GF mice suggest a more dynamic and complex interaction that can be detailed by quantification of brain metabolites in the future. GF mice also exhibit increased striatal turnover of serotonin as evidenced by an increased ratio of 5-HIAA to serotonin in these neurons [62,63].

Growing evidence from GF animal models demonstrates the gut microbiome’s influence on kynurenine degradation pathway. In one study, only male GF mice had elevated circulating TPH levels, and both male and female mice showed a reduced kynurenine to TPH ratio which normalized following transfaunation of GF mice with gut microbes after weaning [64]. Interestingly, despite normalization of gut microbiota, alterations in hippocampal serotonin remained elevated into adulthood suggesting an indelible impact of early life microbial depletion in the hosts.

Investigations in animal models of probiotic administration have provided significant insights into the role of specific microbes in altering systemic tryptophan and serotonergic activity. Other probiotic strains may interact directly with epithelial immune cells to promote differential T cell activation influencing neural activity and neurodevelopment [65]. In this regard, rats given *Bifidobacterium infantis* probiotics had reduced frontal cortical 5-HIAA levels and a marked increase in circulating kynurenine and TPH in association with suppressed inflammation. Further, *Lactobacillus johnsonii* probiotic treatment in rats resulted in decreased kynurenine concentrations and IDO activity in epithelial cells, purportedly via the inhibitory effect of *L. johnsonii*-produced H_2_O_2_, which inhibits IDO by activating its protein oxidation function [66]. Further, epithelial cells may alter microbial communities by production and secretion of bacteriotoxic kynurenine metabolites [6,67].

#### 2.3.2. Indole Pathway

Several bacterial taxa directly utilize TPH by expressing the enzyme tryptophanase, which converts TPH to indole, and these bacteria have been associated with the development of neuropsychiatric disorders, including autism spectrum disorders (ASDs). Roager and Licht provided a comprehensive summary of microbes involved in tryptophan metabolism [68]. Recent work by Kaur and colleagues demonstrate enriched TPH metabolic activity in the five microbial phyla Actinobacteria, Firmicutes, Bacteroidetes, Proteobacteria, and Fusobacteria and the genera of *Clostridium*, *Burkholderia*, *Streptomyces*, *Pseudomonas* and *Bacillus* [69]. Indole and its downstream metabolites indole-3-acetate (ILA) and tryptamine have been shown to attenuate inflammation in intestinal epithelial cells and macrophages [70]. Several other genera, including *Clostridium, Ruminococcus, Blautia and Lactobacillus* expressing TPH decarboxylases catabolize TPH to tryptamine [71]. Microbial tryptophan metabolites such as tryptamine, indole, and indole-3-propionic acid (IPA), in addition to SCFAs, exert various effects by activating or inhibiting intestinal epithelial aryl hydrocarbon receptors (AHRs). AHRs tune adaptive immunity by mediating functional T cell responses, macrophages, and DCs, promote metabolism of xenobiotics, and attenuate lipid metabolism via negative regulation of peroxisome proliferator-activated receptor (PPAR) [72]. Several TPH metabolites can regulate immunity through activating the AHR as well, including *Lactobacillus*-produced indolealdehyde (IAld), which induces IL-22 production and release. Further, IAld and ILA produced by *L. reuteri* augment naïve CD4^+^ T cell differentiation and the induction of regulatory T cells (Tregs) and reprogram epithelial helper T cells to Tregs [73]. Other work shows that ILA can inhibit Th17 cell polarization in mice [74]. Indoleethanol (IE) may also suppress IFNγ-associated activity. ILA has been shown to reduce IL-6 and IL-1β in LPS-activated PBMCs. In addition to immune modulation, TPH catabolites, particularly ILA and IPA, scavenge free radicals and reduce oxidative injury in systemic tissue [75]. Antibiotic ablation of gut microbes or germ-free growth conditions in animal models markedly reduces serum concentration of microbially manufactured protective indole derivatives, such as IPA [76,77]. Evidence suggests that IPA protects against weight gain in animal models and inhibits gut dysbiosis in addition to its putative neuroprotective effect discussed in detail below [78,79]. 

Further studies demonstrated that specific bacterial strains are capable of producing serotonin and other biogenic amines from TPH. These include *Lactococcus lactis cremoris*, *L. lactis lactis*, *Lactobacillus pantarum*, *Streptococcus thermophilus, Escherichia coli* K-12, *Morganella morganii*, *Klebsiella pneumoniae* and *Hafnia alvei* [80,81].

#### 2.3.3. Tryptophan Biosynthesis

While TPH is obtained primarily from the diet in humans, certain gut organisms such as Candida, Streptococcus, Eschirichia and Enterococus express tryptophan synthase that catalyzes the biosynthesis of TPH from serine and indole-3-glycerol-phosphate [4,11]. While some pathogenic bacteria, including *Chlamydia trachomatis* [82], express this enzyme, the prevalence of TPH synthase-expressing bacteria in the human microbiome still remains largely unexplored.

## 3. Tryptophan Metabolites in Neurodevelopment, Neurologic and Psychiatric Disorders

### 3.1. Serotonin and Neurodevelopment

Conversion of tryptophan to serotonin fosters the development and maintenance of the enteric neurons. In rats, development of ECs begins in utero in the duodenum and reaches levels seen in adult rats by birth [83]. Interestingly, once the BBB has been established in developing animals, gut serotonin levels rise dramatically until SERT expression in intestinal epithelium increases in the postnatal period [84]. Enteric nervous tissue develops from invading vagal and sacral neural crest cells that differentiate into neurons and glia. In a critical step in enteric neurogenesis, serotonin activates the 5-HT2B receptors expressed by neural precursors, inducing cell differentiation. Activation of 5-HT4 also impacts enteric neurogenesis. 5-HT4 agonism stimulates enteric neural proliferation, neurite outgrowth, formation of neural networks and development of enteric reflexes [85]. 5-HT4 agonism may also reduce enteric neuronal apoptosis in a CREB-dependent manner [86]. Accordingly, 5-HT4-deficient mice show poor development of enteric neuron density [87]. *Tph2*-deficient mice possess less dopaminergic and GABAergic myenteric neurons compared to controls and diminished intestinal transit activity [88,89]. Gut microbes also influence neurodevelopment, as GF mice develop abnormal neural architecture in the hippocampus and exhibit increased hippocampal neurogenesis [90]. AHRs activated by gut metabolites also stimulate expression of post-synaptic proteins and contribute to hippocampal development [91]. Additionally, in animals and humans, KA has been shown to be critical for maturation of cortical neurons in the perinatal period [92,93].

Serotonin uptake into CNS neurons in the infant occurs in a phasic pattern. Uptake increases in the developing brain in utero and decreases by birth, again rising around 5 weeks of age followed again by normalization. Neuronal SERT expression is highest in childhood and diminishes approximately 1% per year thereafter. Placental *Tph1*-expressing cells have also been shown to play a critical role in brain development through their provision of serotonin to the developing fetus [94].

### 3.2. Serotonin and Cognition

Serotonin is an important neuromodulator in the hippocampus, entorhinal cortex and other brain structures critical for learning and memory. Growing evidence suggests that gut microbes play a central role in central serotonin homeostasis. GF mice, for example, consistently demonstrate difficulty with hippocampal-mediated visual and working memory deficits [90,95]. Cognitive performance in humans can further be enhanced through ingestion of certain probiotic formulations in association with activation of several brain areas which serotonergic neurons influence [96,97].

### 3.3. Neurological Disorders

#### 3.3.1. Alzheimer’s Disease

Compelling evidence indicates a significant influence of TPH metabolites, gut microbes, and associated neuroinflammatory changes in the pathophysiology of Alzheimer’s disease (AD). AD patients have significant differences in gut microbial taxonomy, with decreased Firmicutes and Actinobacteria, and increased Bacteroidetes species compared to age-matched controls [98]. CSF biomarker elevations were correlated with abundance of certain genera, especially *Bacteroides* and *Blautia* [99,100]. In a separate study, AD patients were found to have significantly reduced circulating tryptophan and elevated kynurenine/TPH ratios, which were in turn associated with worse cognitive performance and elevated proinflammatory cytokines [101]. Several preclinical studies analyzing microbiota-produced short chain fatty acids (SCFAs) including butyrate, propionate, and acetate demonstrate protective effects in animal models of AD [102]. TPH metabolites modulate microglial and astrocytic activation in an aryl hydrocarbon receptor-dependent manner [103]. In addition, the mainstays of medical therapy in delaying the progression of AD suggest a plausible role for gut microbes and tryptophan metabolites in development of AD. Memantine, like KA, is an NMDA receptor antagonist thought to reduce amyloid- and inflammation-induced excitotoxicity in neurons. Observation of early loss of cholinergic neurons in the nucleus basalis of Meynert prompted clinical trials to treat AD with acetylcholinesterase inhibitors, which are now the most commonly prescribed medications to prevent AD progression. Studies of acetylcholine-producing gut microbes and lymphocytes may shed further light into the role of microbes in AD pathogenesis. Pocivavsek and colleagues provided circumstantial evidence of the potentially toxic role of kynurenine metabolites in AD, as continuous infusion of luminal kynurenine to mice induced learning and memory deficits in their offspring [104]. Indole pathway metabolites also appear to mediate Alzheimer’s pathogenesis. The indole pathway metabolite IPA inhibits amyloid-β-induced neurotoxicity in vitro and has been developed as a neuroprotective agent for the treatment of AD [105,106].

#### 3.3.2. Parkinson’s Disease

Parkinson’s disease is a progressive neurodegenerative disorder in which aggregates of the protein α-synuclein induce neurotoxicity in the substantia nigra resulting in deficiency in dopaminergic neurotransmission. A wealth of data has now been accumulated implicating the gut microbiome in PD pathogenesis via induction of inflammatory neurotoxicity. Abundance in the gut of the genera *Bacteroides* in these patients correlates with motor symptom severity and levels of proinflammatory TNFα and IFNγ [107,108]. *Verrucomicrobia* in particular are associated with higher levels of circulating IFNγ, illuminating a possible interaction with IDO and TPH metabolites [109]. Recent work has also revealed that tryptophan metabolism is deranged in PD and represents a potential therapeutic target. PD patients have markedly elevated 3-HK in plasma and reduced 3-HANA [110]. KA is significantly reduced and QA levels correlated with disease severity in these patients, indicating a potential pathogenic role of TPH metabolites in exacerbating excitotoxic injury, though a causal role remains to be established. PD patients have elevated kynurenine/TPH ratios in CSF and plasma as well as reduced KAT activity [111]. Consequently, KA synthetic analogues have emerged as neuroprotective drugs of interest for the treatment of PD, HD and AD.

#### 3.3.3. Other Neurodegenerative Disorders

Excess NMDA receptor activation and consequent neuronal excitotoxicity is known to play a role in the pathogenesis of several neurodegenerative diseases. Investigators have postulated that KA may act as an endogenous neuroprotective in its capacity to antagonize excessive NMDA receptor activation. Clinical data suggests a potential role. CSF of patients with end-stage and bulbar-onset amyotrophic lateral sclerosis (ALS) have significantly elevated KA levels compared to controls [112]. In Huntington’s disease patients, free radical-generating 3-HK is highly elevated in early-onset disease, along with striatal and cortical QA. With more advanced disease, however, these concentrations decrease. While KA levels are increased in the CSF of patients with HD [112], post-mortem analysis from HD brains reveal diminished KA concentrations compared to controls, and an elevated serum kynurenine/TPH ratio [113]. Interestingly, the number of glutamine repeats and disease severity negatively correlates with circulating TPH levels. Rat models of HD demonstrate the 3-HK potentiated neural excitotoxicity while free radical scavengers inhibit this effect [114]. Synthetic analogs of KA produce both neuroprotective and antiepileptic effect *in situ* [115,116]. Given the antioxidant effect of IPA, researchers have also proposed the use of this indole derivative for neuroprotection in HD [117].

#### 3.3.4. Multiple Sclerosis

Multiple Sclerosis (MS) is a chronic, progressive and relapsing inflammatory demyelinating disease of the CNS. While the precise inflammatory mechanism underlying disease progression and relapse remains unclear, much evidence suggests a largely B and T cell-driven process. Recently, many medications targeting B and T cell activation have been shown to be clinically efficacious in relapse prevention [118]. Microbiota analysis of MS patient stool samples revealed increases in *Methanobrevibacter* and *Akkermansia* and decreased *Butyricimonas* in MS patients compared to controls [119]. While the impact of these differences in MS pathogenesis is unclear, gut microbial metabolites may contribute to MS pathogenesis. TPH metabolites and type I IFN signaling has been shown to activate astrocytic AHRs and thereby suppress CNS inflammation in experimental allergic encephalomyelitis (EAE) models of MS. Patients with MS also have reduced circulation of TPH metabolite AHR agonists, implicating TPH metabolic dysregulation in MS pathogenesis [120]. Research evaluating the kynurenine pathway in this disorder has yielded interesting outcomes. Patients suffering relapse have elevated KA levels, while post-mortem samples show reduced KAT activity [120]. Further, QA may induce oligodendrocyte apoptosis contributing to demyelinating injury [121]. In other EAE models, data demonstrated central accumulation of toxic kynurenine metabolites. Gut microbes also influence central immunity as immune overactivation secondary to microbial changes exacerbates induced inflammatory injury in these models [122].

#### 3.3.5. Cerebrovascular Disease

TPH metabolites may influence the development and severity of several types of cerebrovascular disease, both in disease onset and in contribution to other known cerebrovascular disease risk factors. Interest in the role of inflammation induced by acute ischemic stroke (AIS) has led to studies investigating TPH metabolism in these patients. Evaluation of TPH metabolite concentration in the blood of patients with AIS, significant carotid artery stenosis and controls demonstrated that individuals with carotid stenosis and AIS showed lower circulating TPH and 3-HAA, and higher circulating arachidonic acid (AA) and 3-HK [123]. Another study in stroke patients showed a positive correlation between the kynurenine/TPH ratio and stroke severity [124]. In AIS patients, levels of the oxidative stress marker malondialdehyde (MDA) were inversely related to concentration of KA [125]. Several polymorphisms in genes relevant to TPH metabolic pathways were significantly higher in AIS patients compared to controls [126]. Metabolomic profiling of AIS patients demonstrated elevated serum lactate, carbonate and glutamate, and lower levels of TPH in addition to several other amino acids [124]. 

The gut microbiome has similarly been studied in stroke patients and in animal models. Data demonstrated that microbial diversity collapsed after AIS in the days following stroke onset [127]. Microglia were major effectors of inflammatory injury encountered after stroke [128]. Preclinical stroke models demonstrated M1 polarization of microglia following a short period of predominantly M2 polarization, with activation of astrocytes and recruitment of peripheral macrophages [129]. 

Gut microbiota control maturation and function of microglia throughout life through SCFA production, vagal transit, or by production of other metabolites that cross the BBB. Microbial TPH metabolites influence microglial function by modulation of TGFα and VEGFβ in an AHR-mediated process [130]. Some evidence suggests that commensal microbes may exacerbate the response to AIS. In one middle cerebral artery occlusion (MCAo) model, GF mice had significantly lower stroke burden than conventional mice [131]. Premorbid gut dysbiosis confers additional risk by contributing to the pathogenesis of well-established stroke risk factors, including diabetes mellitus, hypertension, obesity and metabolic syndrome [132,133]. Further work to elucidate differential mechanisms by which gut microbes influence stroke etiology may further provide required insight into the identification of therapeutic targets in the future.

Development of other types of cerebrovascular disease may influence, or be influenced by microbial TPH metabolites. Subarachnoid hemorrhage (SAH) occurs when dilatation of large intracranial arteries leads to aneurysm formation and rupture in the subarachnoid space. In severe disease, patients can develop vasospasm, causing delayed cerebral ischemia several days after ictus. Antibiotic depletion of gut microbes protected against aneurysm formation in a hypertension/elastase model of intracranial aneurysm formation [134]. Experimental SAH data demonstrated a marked increase of serotonin in intracranial arteries [135]. Networks of serotonergic pial perivascular nerves develop after SAH, indicating a role of serotonin accumulation in vasospasm pathogenesis [136]. Notably, in five patients admitted to a neurocritical care unit for SAH, lumbar drain samples were analyzed for vasogenic amines. All patients had elevated CSF serotonin, but these were highest in patients who developed vasospasm. Interestingly, tryptophan levels spiked with the development of vasospasm. Such findings indicate that gut microbial modulation that reduces central tryptophan availability may decrease the incidence and severity of vasospasm, however, this hypothesis remains to be tested. A related disorder, reversible cerebral vasoconstriction syndrome (RCVS), is known to be caused by agents that increase central serotonin levels, including SSRIs and triptans. One case successfully responded to the serotonin antagonist cyproheptadine [137], though additional work is required to further elucidate the role of TPH availability, gut microbiota and RCVS.

#### 3.3.6. Migraine

Migraine is a very common headache disorder, in which the gut microbiome and TPH metabolism may play a key role. Migraine frequently occurs as a comorbid condition with other disorders in which dysfunctional TPH metabolism and gut metabolites may be implicated in the disease pathogenesis including IBS, anxiety and depression [138,139]. While studies continue to piece together the precise mechanisms underlying migraine pathogenesis, serotonin and calcitonin gene-related peptide (CGRP) significantly impact trigeminal, hypothalamic, thalamic activation as well as meningeal vasodilation. Functional MRI data indicate that brain functional connectivity is pathologically altered in individuals with migraine [140]. The role of the gut microbiome and associated TPH metabolites in neurodevelopment may play an early role in conferring susceptibility to migraine development. Triptans, serotonin agonists, are the first-line therapeutic agents for abortive treatment of migraine. It has been postulated that the vasoconstrictive effect of serotonin agonism on meningeal vessels mediates the therapeutic effect of these medications. Thus, a mechanistic link between TPH availability and serotonin production mediated by gut microbes may contribute to acute exacerbations of migraine. Excess excitatory glutamatergic neurotransmission is also felt to be a characteristic feature of migraine pathophysiology, one that may theoretically be modulated by relative KA and QA availability [141]. Clinical studies focusing on gut microbial changes in migraine patients have also shown that dysbiosis can worsen migraine pain in a TNFα-dependent process. Analysis of fecal samples of elderly women with migraine revealed significant elevation of Firmicutes, especially *Clostridium* species, in migraineurs compared to age-matched controls [142]. Kynurenine metabolites were also elevated in this group of patients. Interestingly, probiotic formulations in clinical trials have generally shown positive outcomes [143]. Thus, larger scale clinical trials and identification of beneficial gut microbial strains may provide new therapeutic strategies to treat patients affected by migraine. 

#### 3.3.7. Traumatic Brain Injury

Finally, traumatic brain injury (TBI) is a “silent epidemic” that induces initial trauma-related brain injury followed by inflammation-driven secondary brain injury which peaks days after the initial trauma and can last for years thereafter [144]. Release of damage-associated molecular patterns (DAMPs) by injured cells promote microglial activation and recruitment of peripheral proinflammatory cells to the CNS exacerbating cerebral edema, ischemia and free radical injury [145]. Although the impact of premorbid dysbiosis on severity of injury after TBI remains to be elucidated, TBI itself induces dysbiosis that can last for years [146]. A study employing a rabbit pediatric TBI model revealed TBI induced IDO upregulation around the area of injury that persisted for weeks following induced trauma, as well as a significant increase in circulating kynurenine one week after injury [147]. Further, serotonin/TPH and melatonin/TPH ratios were significantly decreased three weeks after trauma, suggesting dysregulated TPH metabolism as a consequence of TBI [148]. Cytokine profiles correspondingly showed significant elevations in proinflammatory TNFα, IL-6 and IL-1β after initial injury followed by increased regulatory cytokines IL-10, IL-4 and TGFβ days after the proinflammatory response in these patients [149,150]. Studies collecting CSF from patients with severe TBI demonstrated gradual increase in CSF TPH in the first three days of injury that subsequently normalized. Interestingly, QA became markedly elevated by day 3 after trauma [150]. Nonetheless, further studies are necessary to potentially link gut microbial TPH metabolism with secondary brain injury following TBI.

#### 3.3.8. Minocycline in Brain Injury

Minocycline is a neuroprotective antibiotic that has been studied clinically in multiple disease models, including acute traumatic brain injury, ischemic stroke, and various neurodegenerative disorders. Its putative function involves microglial inactivation and other immunoregulatory mechanisms [151]. It has also been noted to inhibit IDO activity indirectly, though its impact on the gut microbiome in its capacity as a broad-spectrum antibiotic is yet unclear [152].

### 3.4. Psychiatric Disorders

#### 3.4.1. Anxiety and Depression

Serotonergic signaling plays a crucial role in the pathogenesis of several neurologic and psychiatric disorders.

Tryptophan metabolites, especially serotonin, are centrally implicated in the pathogenesis of anxiety and depression. Medications that promote central serotonin availability, especially selective serotonin reuptake inhibitors (SSRIs), MAO inhibitors (MAOIs), and tricyclic antidepressants (TCAs) have revolutionized the treatment of these disorders. With the increase in understanding of the gut microbiome constitution in different disease processes, it has become clear that gut microbes may play a crucial role in the nascence and clinical phenotype of these disorders. GF mice display more anxious behavior than conventionally-raised mice, behavior that is not readily reversible with microbial repopulation after weaning in the hosts, suggesting a critical period during which the gut microbiome may support psychological development [153]. Furthermore, several probiotics have shown efficacy in reducing anxiety and depression in animal models and humans. TPH supplementation reduces anxiety, possibly by providing adequate substrate for conversion to serotonin; nonetheless, this remains controversial. Excessive TPH is found in GF mice in association with anxiety behavior, suggesting a more complex pathophysiology in anxiety and depression than being solely driven by TPH deficiency [153]. Probiotics that included species such as *Lactobacillus helveticus* R0052 and *Bifidobacterium longum* R0175 reduced anxiety, enhanced emotional well-being, and reduced depressive symptoms in mice and humans, respectively [154].

Mice with fecal microbial transplant from depressed patients demonstrate worse anxiety in association with higher circulating kynurenine and kynurenine/TPH ratios [155,156]. Interestingly, chronic stress is known to increase circulating TPH and cortisol, resulting in shunting of serotonin metabolism toward kynurenine and its metabolites due to enhanced glucocorticoid-induced TDO expression [157]. Findings from induction of stress in mice showed that exogenous butyrate regulated stress-induced depressive behavior, decreased hippocampal serotonin and increased hippocampal brain-derived neurotrophic factor (BDNF) [158,159]. Mice whose gut microbiota was depleted with antibiotic cocktails demonstrated anxiety-like behaviors and elevated circulating kynurenine [156]. In obese rats fed a high-fat diet, anthocyanins protected against neuroinflammation and also exhibited reduced circulating TPH and increases in KA [160].

#### 3.4.2. Schizophrenia

Conversion of tryptophan to serotonin may be impaired in the pathogenesis of schizophrenia, as certain *TPH1* polymorphisms increase susceptibility to schizophrenia and suicidality [161]. Further, low 5-HIAA levels in the cerebrospinal fluid (CSF) were associated with suicidality and aggressive behavior [162]. Kynurenine metabolites may also play a causative role, as cortical KA levels are increased in schizophrenic patients [163]. Studies in animal models indicate that TPH suppresses aggressive behavior, likely related to increased central serotonin availability [164,165]. Sekar and colleagues [166] published a landmark genome-wide association study that identified genetic loci implicated in the pathogenesis in schizophrenia, which involves complement C4-mediated microglial synaptic pruning overactivation. Given the established role of the gut microbiome in mediating central immunity and the preponderance of case-control studies demonstrating gut dysbiosis in schizophrenic patients, investigators have sought to link dysbiosis with immune dysregulation leading to overactive synaptic pruning during critical periods of brain development. Epidemiological studies also support the hypothesis that maternal immune activation induced by systemic infection is an independent risk factor for development of schizophrenia in offspring [167]. One recent study indicates that prenatal immune exposure resulted in upregulation of frontal cortical C4 activity [168]. The vertical transfer of the maternal microbiome to the offspring may also lead to sustained immune dysfunction and increased risk for synaptic over-pruning. Further mechanistic work is required to understand the interactions of gut microbes, TPH metabolites and the host immunity in the pathogenesis of schizophrenia and other neurodevelopmental disorders.

#### 3.4.3. Autism Spectrum Disorders

Patients with autism spectrum disorders (ASD) may be deficient in tryptophan per clinical studies of circulating TPH and examination of excreted kynurenine metabolites [169]. Metabolomic analysis of lymphoblastoid cells derived from ASD patients showed reduced NADH generation when tryptophan was the sole available energy source, suggesting potential impairment in the QA degradation pathway in these patients [170]. Certain microbial species may be involved in the pathogenesis of this disease. Accordingly, several studies evaluating differential gut microbial abundance in ASD patients linked autistic symptoms with lower abundance of *Prevotella*, *Coprococcus* and *Veillonellaceae. Bacteroides fragilis*, a tryptophanase-synthesizing bacterium, may reduce TPH availability in patients with ASD [171,172]. Non-TPH-derived microbial metabolites may also play a causal role, as one study observing gut microbial metabolites from a mouse maternal immune activation (MIA) model of ASD showed a 46-fold increase in microbial metabolite 4-ethylphenylsulfate, which normalized if mice were colonized with *B. Fragilis* [173].

### 3.5. Gastrointestinal Disorders

#### 3.5.1. Irritable Bowel Syndrome

Irritable Bowel Syndrome (IBS) is characterized by abdominal symptomatology without a clear organic etiology. These patients experience low pain thresholds and visceral hypersensitivity, as well as diarrhea, constipation or both. IBS patients demonstrate variable SERT expression depending on the clinical phenotype [174]. Whereas those with diarrhea-predominant IBS (IBS-D) have elevated circulating serotonin levels putatively due to reduced uptake into platelets, those with constipation-predominant IBS (IBS-C) have reduced circulating levels and increased platelet serotonin concentration [175]. Experimental alteration of serotonin availability in mice with prolonged treatment with selective serotonin reuptake inhibitors (SSRIs) causes delayed GI transit and stool output [19]. Chronic SSRI use in humans similarly reduced peristaltic activity, once again demonstrating differential effects based on dose duration. Variability in the single nucleotide polymorphism rs25531 in the gene encoding SERT appears to confer significantly elevated risk of IBS and depression, reinforcing the hypothesis that serotonin regulation potentially drives the pathogenesis of IBS [176].

Metabolism of tryptophan also appears to be altered in these patients, as IDO activity is elevated [177]. Correspondingly, circulating kynurenine and kynurenine/TPH ratios are elevated in patients with IBS [178]. Responsiveness of IDO to immune activation has prompted investigations in patients and animal models, which revealed alteration in TLR expression in IBS patients. Further analysis of patients’ serum suggests that activation of different TLRs induced differential induction of tryptophan metabolism via the kynurenine degradation pathway [179].

Patients with IBS experience visceral hypersensitivity as a cardinal symptom of the disease process. Neuroimaging studies in IBS patients show alterations in brain regions important in central processing of pain, in which serotonin signaling is known to be influential [180]. Increasing available central serotonin with antidepressant TCAs and SSRIs improve symptoms of IBS. Further, agonism of 5-HT3 receptors on afferent spinal nerves activates neural transmission in the dorsal horn potentiating responsiveness to pain [181]. 5-HT3 antagonism inhibits dorsal horn activation following colonic distension [182]. In rats, early life antibiotic-driven depletion of gut microbiota is associated with the development of visceral pain as adults that is ameliorated with early administration of probiotics [183]. Further, certain gut microbes can either exacerbate, or improve GI symptoms, though the link between gut microbes and tryptophan metabolites still requires further investigation.

#### 3.5.2. Inflammatory Bowel Disease

Intestinal mucosal injury incurred in the setting of inflammatory bowel diseases such as Crohn’s, ulcerative colitis and celiac disease can pathologically alter tryptophan metabolism and serotonin synthesis. These disorders induce differential changes in serotonin conversion and EC abundance. Accordingly, patients with ulcerative colitis have reduced mucosal ECs and serotonin levels, whereas patients with celiac disease have increased circulating serotonin and ECs [184,185]. Though, SERT levels are consistently lower than controls in each disease. Preclinical models of GI infection or inflammatory disease almost uniformly demonstrate elevated serotonin release and EC density as well as decreased epithelial SERT expression, suggesting reduced EC density and serotonin in patients with UC may be reflective of protective downregulation or sequelae of epithelial injury. In epithelial cell culture models, treatment with inflammatory cytokines IFNγ and TNFα cause a dose-dependent suppression of SERT expression and serotonin uptake [186]. Investigations in murine models of induced colitis also demonstrate increased sensitivity to serotonin in vagal afferents [187]. Further implicating serotonin in the pathogenesis of inflammatory GI disorders, *TPH1*-deficient mice have a decreased severity of colitis [188]. GI infections have also been shown to increase TDO and IDO activity with associated increased kynurenine-to-TPH ratios [61]. These findings illuminate a potential mechanism by which gastrointestinal diseases can increase circulating serotonin and induce hypercoagulability, rendering afflicted individuals susceptible to thrombotic neurologic disorders. Indeed, several categories of gastrointestinal disorders are associated with increased risk of future ischemic stroke [189]. 

#### 3.5.3. Age-Related Gastrointestinal Dysfunction

Advanced age increases individuals’ susceptibility for GI dysfunction which has been attributed to age-related neuronal loss. 5-HT4 agonists, which stimulate neurite growth and network formation in the developing intestine have also been shown to protect against neuronal apoptosis and inflammation-induced axonal degeneration and autophagy. Additionally, 5-HT4 receptor agonism promotes adult enteric neurogenesis [87]. Correspondingly, otherwise healthy individuals of advanced age show reduced circulating tryptophan, potentially limiting serotonin availability.

## 4. Summary and Conclusions

TPH and its metabolites are involved in a host of physiologic and pathologic processes. Dietary tryptophan is absorbed in the small intestine, where it can be transported to the peripheral circulation, be converted to serotonin by enterochromaffin cells, or catabolized via the kynurenine degradation pathway in peripheral tissues. Kynurenine degradation proceeds to produce either neuroprotective kynurenic acid or neurotoxic quinolinic acid. Unabsorbed TPH serves as a substrate for gut microbial degradation, where microbial metabolites may influence systemic inflammation. Serotonergic neurons and glia take up TPH and convert it to serotonin, which is a critical neurotransmitter in limbic processing of emotion, learning and memory. TPH availability also drives enteric and central neurodevelopment through serotonin activity. Pathologic states may influence, or be driven by TPH and its many bioactive metabolites. These metabolites may foster premorbid risk factors and impact the natural history of many diseases, including inflammatory bowel syndrome, neurodegenerative disorders, neurodevelopmental disorders, and cerebrovascular disorders. Thus, understanding the TPH metabolic pathway in these diseases may shed light on potential future therapeutic targets (Figure 2). Further, our growing understanding of TPH physiology and its metabolites suggest that gut microbe-dependent metabolism of TPH may be involved in the pathogenesis of numerous central disease processes. Hence, identification of pathogenic and beneficial bacteria and their mechanistic linkage to TPH metabolism may enable precision-based approaches to treat a wide array of neurological and psychiatric disorders. While much investigation is underway, there exist major opportunities to elucidate the mechanistic underpinnings of these disease processes and thereby improve the quality of life of those suffering from these disorders.

## Figures and Tables

**Figure 1 ijms-22-02973-f001:**
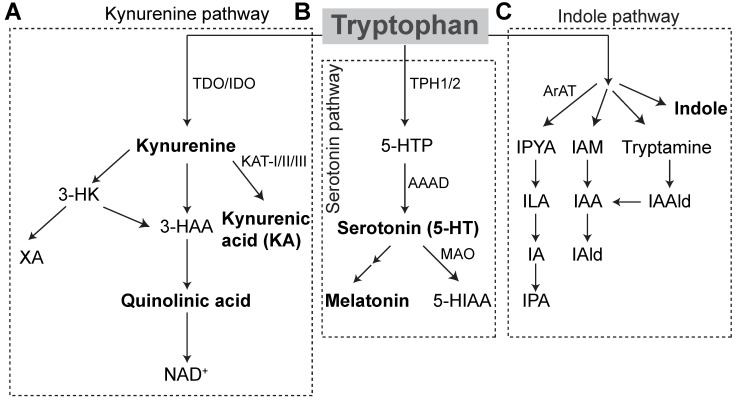
Tryptophan Metabolic Pathways. Tryptophan metabolism in hosts occurs via the kynurenine pathway or the serotonin pathway to produce bioactive metabolites. (**A**) 90% of kynurenine pathway degradation occurs in the liver via TDO conversion of TPH to kynurenine. The remaining kynurenine degradation occurs by IDO in the brain, GI tract and liver. Kynurenine is metabolized to kynurenic acid by KAT enzymes. It can also be metabolized to quinolinic acid, which is then converted to NAD^+^. Alternatively, kynurenine is converted to 3-HK and subsequently to XA. (**B**) In the serotonin pathway, TPH is converted to 5-HTP by TPH1 in enterochromaffin cells, or TPH2 in enteric or central neurons. 5-HTP is decarboxylated to form serotonin. Serotonin can be further metabolized to form melatonin, or degraded by MAO to 5-HIAA, which is excreted in the urine. (**C**) In gut microbes, tryptophan is metabolized into indole and indole derivatives. Microbes express different enzymes that utilize TPH. Conversion of TPH to IPA occurs initially through aromatic amino acid decarboxylase to IPYA, which is then converted to ILA and IA prior to conversion to IPA. Through other tryptophan-degrading enzymes, microbes can create metabolic end-products IAld and IAAld. IAAld, which is formed from tryptamine, can also be converted to IAA and subsequently to IAld. Microbes may also convert TPH directly to indole. AAAD = aromatic amino acid decarboxylase, ArAT = aromatic amino acid aminotransferase, 3-HAA = 3-hydroxyanthranilic acid, 5-HIAA = 5-hydroxyindoleacetic acid, 3-HK = 3-hydroxykynurenine, 5-HT = 5-hydroxytryptamine, 5-HTP = 5-hydroxytryptophan, IA = anholocyclic acid, IAA = indole-3-acetic acid, IAAld = indole-3-acetaldehyde, IAld = indole-3-aldehyde, IAM = indole-3-acetamide, IDO = indoleamine 2,3-dioxygenase, ILA = indole-3-lactic acid, IPA = indole-3-propionic acid, IPYA = indole-3-pyurvic acid, KAT = kynurenine aminotransferase, MAO = monoamine oxidase, NAD = Nicotinamide adenine dinucleotide, TDO = tryptophan 2,3-dioxygenase, XA = xanthurenic acid.

**Figure 2 ijms-22-02973-f002:**
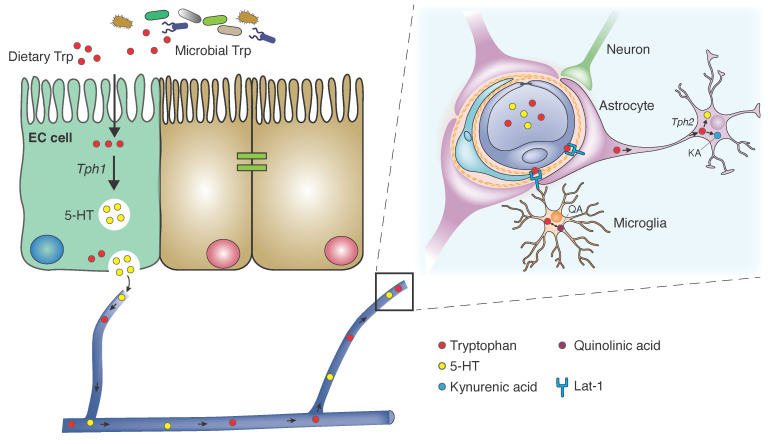
Tryptophan Metabolism and Transport. Dietary and microbial tryptophan (TPH) is absorbed in small intestinal epithelium. In enterochromaffin cells (ECs), TPH is released unaltered into the peripheral circulation, where it is degraded in the liver and other peripheral tissues to kynurenine and its metabolites. Or, it is converted to serotonin in ECs via tryptophan hydroxylase 1 (TPH1). TPH is transported across the blood-brain barrier (BBB) into astrocytes via Lat-1 large neutral amino acid transporter. In neurons and glia, it is converted to serotonin via TPH2, or catabolized by the kynurenine degradation pathway to kynurenic acid. Most of the brain quinolinic acid derives from TPH catabolism in microglia.

## Data Availability

No new data were created or analyzed in this study. Thus, data sharing is not applicable to this article.

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
