# Peer review of "Tryptophan Metabolism and Gut-Brain Homeostasis"

_ijms, 2021, doi:10.3390/ijms22062973_

Round 1

Reviewer 1 Report

Review submitted to IJMS titled: Tryptophan Metabolism and the Gut Microbiota-Brain Axis, describes very specific and interesting topic related to tryptophan - very well known and described in literature amino acids. Authors, decided discuss about absorption and degradation of  tryptophan, and , what is interesting, authors focused into manuscript Kynurenine Degradation Pathway as well as Gut Microbial Tryptophan Metabolism and they brought these facts together in describing Neuro- and Psychopathology.

MAnuscript is written in correct way, and it could be published in present form. 

Reviewer 2 Report

Authors of the presented article entitled: "Tryptophan Metabolism and the Gut Microbiota-Brain Axis" evaluated interesting subject of correlations between pathways of tryptophan metabolism, actions of its metabolites and potential role in pathophysiology of psychiatric and neurological disorders.

Broad comments

It is a well written review assesing an interesting topic of complex metabolism of a tryptophan, an exogenous amino acid. Additionally, broad number of neurological and psychiatric disorders have been evaluated what is definitelly an asset of this review. Depsite these positive aspects there are many concerns in presented manuscript that I will address below.

  1. Authors submitted article in file without numerical order of lines and pages making it more difficult for reviewers to analyze and address suggestions.
  2. The title is a little bit misleading since authors focused mainly on serotonin pathway and kynurenine pathway and only in some places mentioned formation and effects of gut bacteria-derived metabolites of TRP including indole etc. The main text, running title, conclusions and Figure 1 suggest limited role of those metabolites in presented article, however, main title suggests something else. Authours should think about this aspect.
  3. In general the article is not well-organized. There should be more subunits and paragraphs. At the beginning there are some, but later on there is one Neuro- and Psychopathology that is for 10 pages and includes evaluation of psychiatric, neurological and also intestinal disorders (UC, celiac disease). It is really lacking any sense. Additionally within this paragraph authors evaluate anxiety, depression, than ALS, Huntington, Alzheimer diseases than again psychatric like shizophrenia. Authors should stick to neurological and psychiatric order or divide this paragraph into 2 and separately review neurological and psychiatric diseases.
  4. Authors did not evaluate sufficiently role of gut microbiota-derived metabolites in presented diseases. Indole-3-propionic acid is known to decrease body mass, not only serotonin. It should be discussed. Additionally, this metabolite was proven to reveal protective effect against b-amyloid proteins in vitro and its concentration was also assessed in patients with Huntington disease.
  5. When it comes to serotonin I would suggest including a sentence evaluating role of 5-HT3 antagonists as aniemetic medications in paragraph on page 5 lines 3-10.
  6. In a few lines of presented article authors jump too fast from reviewing effects of tryptophan metabolism to conclusions that these efects are related to gut bacteria metabolism. There is gut-brain axis and also there are correlations assesing activity of gut bacteria not only endogenous host's cells present in intestines. For example on the bottom of page 19 authors wrote: "All patients had elevated CSF serotonin, but these were highest in patients who developed vasospasm. Interestingly, tryptophan levels spiked with the development of vasospasm. Such findings strongly suggest a potential role for the gut microbiome in SAH and vasospasm." It does not prove that since serotonin in intestines if formed mostly by enterochromatofine cells and not gut bacteria. Authors should divide these two aspects or prove by reviewing scientific literature that there are really those associations. It should be revised. 
  7. On page 12 lines 17-27 authors evaluated impact of antibiotics on tryptophan metabolism. One article from 2019 evaluated changes in concentration of  gut microbiota-derived metabolites of tryptophan (including indole-3-propionic acid)  of tryptophan after antibiotic treatment and might be worth discussing. 
  8.  I would suggest adding additional figure showing complex metabolism of tryptophan and 3 main pathways of formation of its metabolites. Additionally a table evaluating specific actions of serotonin depending on specific tissues toghether with specific receptors taking part in those effect would be helpful to better understand complex biological effects of serotonin. 

Specific comments

  1. there is "brain stem" and later on "brainstem" on page 6 line 20
  2. Authors miswrote name of indole-3-propionic acid on page 10 line 7, additionally since it's the first time this metabolite appears in the text abbrev. should follow this place and should not be introduced on page 10 line 17.

Round 2

Reviewer 2 Report

Authors submitted revised version of article entitled: "Tryptophan Metabolisms and Gut-Brain Homeostasis". I believe that corrected title of the article gives more insight into scientific topic presented by authors.

Article was revised according to reviewer's suggestions. Corrections introduced by authors improved its value and made it more organized and easier to read. Interesting aspects concering role of gut bacteria-derived metabolites of tryptophan have been included showing broad spectrum of biological functions affected by metabolism of tryptophan.  

I believe that in present form this article is a valuable review summarizing biological effects of tryptophan and its metabolites in neurological, psychical and gastrointestinal homeostasis. 

I have only one suggestion. Figure 1 evaluates complex metabolism of tryptophan. Maybe by dividing it into 3 descriptions (A,B,C) it would be more clear for the readers. However, it is great that authors implemented this figure and additional table in revised version of the manuscript. 
